# Impact of Obesity and Diabetes in Pregnant Women on Their Immunity and Vaccination

**DOI:** 10.3390/vaccines11071247

**Published:** 2023-07-17

**Authors:** Magdalena Wierzchowska-Opoka, Arkadiusz Grunwald, Anna K. Rekowska, Aleksandra Łomża, Julia Mekler, Miracle Santiago, Zuzanna Kabała, Żaneta Kimber-Trojnar, Bożena Leszczyńska-Gorzelak

**Affiliations:** Chair and Department of Obstetrics and Perinatology, Medical University of Lublin, 20-059 Lublin, Poland; magdaopoka11@gmail.com (M.W.-O.); arek.grunwald.2@wp.pl (A.G.); arekowska@icloud.com (A.K.R.); aleksandra.lomza98@gmail.com (A.Ł.); julia.mekler@yahoo.com (J.M.); msantiago.mul@gmail.com (M.S.); zuzanna.kabala00@gmail.com (Z.K.); bozena.leszczynska-gorzelak@umlub.pl (B.L.-G.)

**Keywords:** pregnancy, obesity, diabetes, immunity, vaccination, COVID-19

## Abstract

Pregnant women with obesity and diabetes are at increased risk of developing infections and other complications during pregnancy. Several mechanisms are involved in the immunological mechanisms that contribute to reduced immunity in these populations. Both obesity and diabetes are associated with chronic low-grade inflammation that can lead to an overactive immune response. Pregnant women with obesity and diabetes often have an increase in pro-inflammatory cytokines and adipokines, such as TNF-α, IL-6, IL-1β, leptin, and resistin, which are involved in the inflammatory response. Insulin resistance can also affect the functioning of immune cells. Furthermore, both conditions alter the composition of the gut microbiome, which produces a variety of biomolecules, including short-chain fatty acids, lipopolysaccharides, and other metabolites. These substances may contribute to immune dysfunction. In addition to increasing the risk of infections, obesity and diabetes can also affect the efficacy of vaccinations in pregnant women. Pregnant women with obesity and diabetes are at increased risk of developing severe illness and complications from COVID-19, but COVID-19 vaccination may help protect them and their fetuses from infection and its associated risks. Since both obesity and diabetes classify a pregnancy as high risk, it is important to elucidate the impact of these diseases on immunity and vaccination during pregnancy. Research examining the efficacy of the COVID-19 vaccine in a high-risk pregnant population should be of particular value to obstetricians whose patients are hesitant to vaccinate during pregnancy. Further research is needed to better understand these mechanisms and to develop effective interventions to improve immune function in these populations.

## 1. Introduction

Myriad physiologic changes occur during pregnancy, including numerous alterations that affect the immune system. The hormonal changes experienced during gestation resonate throughout the body, impacting various organ systems [1].

Perhaps the most far reaching of all physiologic changes in pregnancy is the smooth muscle relaxation caused by increased progesterone, which decreases systemic vascular resistance (SVR) and arterial blood pressure. Venous return and cardiac output increase dramatically—up to 60% during the first trimester—in order to meet increased metabolic demand. Decreased SVR causes renal vasodilation, causing a 50% increase in glomerular filtration rate (GFR). As compensation for this decreased SVR and blood pressure, the renin angiotensin aldosterone system is continuously stimulated, causing a marked increase in aldosterone levels. Decreased intestinal tone of the stomach and gastroesophageal sphincter causes reflux and prolonged gastric emptying, while decreased tone in the intestines increases water reabsorption and constipation [1].

Liver metabolism increases in pregnancy, which, when combined with a decrease in lipoprotein lipase activity, causes an increase in plasma triglycerides and total serum cholesterol; though both low-density lipoprotein (LDL) and high-density lipoprotein (HDL) increase during pregnancy, LDL increases more markedly, and thus total cholesterol ratio increases. Human placental lactogen (hPL) increases lipolysis and free fatty acids, acting as an insulin antagonist to induce a diabetogenic state, causing hyperplasia of the islet cells and increased insulin secretion. A typical pregnant patient is insulin sensitive in the first trimester, becomes insulin resistant in the second, and maintains her insulin resistance till term. Placental corticotropin-releasing hormone (CRT) causes as much as a three-fold increase in plasma cortisol levels, resulting in a hypercortisol state [1].

Hematologic changes also occur; while both plasma volume and erythrocyte volume increase, there is a more marked increase in plasma, resulting in decreased hematocrit and dilution anemia. Increased venous stasis heightens the pregnant patient’s risk of thromboembolism [1].

Adaptations within the maternal immune system are the sites of many physiologic changes that act to protect the mother and fetus from pathogens. This increased immune responsiveness must strike a balance in order to maintain a healthy pregnancy, providing protection while avoiding detrimental immune response against the allogenic fetus. Although the precise mechanism is unclear, research supports a dynamic cooperation between the maternal and fetal immune systems, as opposed to a broad maternal immune suppression [2].

Obesity fosters a state of chronic, systemic inflammation, increasing the risk of myriad comorbidities, including nonalcoholic fatty liver disease, retinopathy, cardiovascular disease, nephropathy, and autoimmune disease (e.g., rheumatoid arthritis). Obesity confers insulin resistance, often leading to type 2 diabetes mellitus (T2DM); chronic tissue inflammation has been observed in insulin target tissues—e.g., adipose, liver, muscle, pancreatic islets—and is thus understood to be a key component of the diabetes disease state. The driving force of this chronic low-grade inflammation is the recruitment, accumulation, and activation of pro-inflammatory macrophages, though other immune markers also contribute to this process, e.g., tumor necrosis factor-α (TNF-α), which is over-expressed in adipose tissue, and activates various inflammatory signaling molecules, such as c-Jun N-terminal kinase (JNK) and inhibitor κB kinase beta (IKKβ). In addition to TNF-α, other adipokines contribute to the chronic inflammatory response, such as interleukin 1β (IL-1β), which is activated downstream from an “inflammasome” (a cytosolic protein complex), activated by excess glucose and free fatty acids, and impairs adipocyte insulin signaling and leptin, which can enhance steatosis and contribute to hepatic inflammation. The role of the immune system in T2DM is so inextricably linked to the co-morbid changes in glucose metabolism that scientists have coined the term “immunometabolism” to refer to this interplay of systems [3].

A quarter of all pregnancy-related complications—e.g., gestational diabetes mellitus (GDM), gestational hypertension, preterm labor, macrosomia—are associated with maternal obesity. Excess adipose tissue acts as an active endocrine organ, causing numerous deleterious effects throughout the body, including dysregulation of the metabolic, vascular, and inflammatory pathways, which can affect obstetric outcomes [4]. Obesity and diabetes mellitus (DM) are, both separately and in combination, criteria for high-risk pregnancies, with the comorbidity of obesity and DM resulting in a significant increase in obstetric complications [4].

Pregnancy is associated with increased susceptibility to infection and increased severity of infection, as well as adverse fetal outcomes associated with specific microorganisms. Pregnancy confers a decrease in B-l cells, which causes a deficit of IgM antibodies, as well as a decrease in the functionality of B cells; these changes weaken the pregnant patient’s innate immunity and increase her susceptibility to infection. Activity of the complement system increases during pregnancy, increasing the severity of the immune response to specific antibodies (e.g., influenza). Greater increases in complement activity are associated with various obstetric complications (e.g., preeclampsia) [2].

Alterations in immunity during pregnancy, in conjunction with the vulnerability of the fetus, can lead to various severe adverse fetal outcomes. Risk of infection increases due to maternal alterations in immunity; pregnancy has been described as a state of “up-regulated innate immune response and decreased cell-mediated response” [5], which is maintained via a change in cytokine phenotype (i.e., a switch from Th1- to Th2-predominance). This observed decrease in cell-mediated immunity increases the likelihood of infection [5]. TORCH infection—in utero exposure to specific pathogens associated with syndromic congenital defects—occurs via vertical transmission, when maternal microorganisms cross the placenta. While the mechanisms of congenital transmission are not fully elucidated, it is understood that pro-inflammatory cytokines trigger an immune response and result in tissue damage. Typically, TORCH infection (e.g., *Toxoplasmosis gondii*, rubella, Zika) occurs due to acute, primary infection during pregnancy, though pre-gestational and chronic infections can also be reactivated. Risk of TORCH increases with gestational age, due to changes in maternal immune factors in addition to the decreasing anatomical thickness of the placenta; in contrast, infection is more severe, and clinical manifestation more marked, when exposure occurs earlier in the pregnancy, and maternal comorbidities also impact clinical severity [6]. The severity of complications is also influenced by co-infection with TORCH pathogens [7].

Both obesity and T2DM are associated with an increased risk of recurrent, nosocomial, and secondary infections, with higher rates of sepsis and death. Patients present a higher risk of developing cutaneous infections, catheter–related infections, and various categories of pneumonia (e.g., community-acquired, ventilator-associated, aspiration), with worse prognoses. Though the biomolecular causes are not been fully understood, research has indicated that the neutrophils of obese patients demonstrate certain characteristics (e.g., increased chemotaxis and random migration, increased release of basal superoxide) that suggest they are chronically primed to fight infection, while demonstrating poorer efficiency to do so when compared to normal-weight patients; neutrophils in T2DM patients show impairment as well, though across nearly all functions (e.g., migration to inflammatory sites, release of lytic proteases, phagocytosis, production of reactive oxygen species, apoptosis) [8]. Wound healing is delayed in DM, and diabetic wounds, including pressure ulcers due to immobilization, represent a major comorbidity. Impairment of wound healing occurs due to overactivity of pro-inflammatory cytokine pathways; innate immune cells are activated by pathogen-associated molecular patterns (PAMAs), as well as damage-associated molecular patterns (DAMPs), which trigger and maintain an inflammatory state [9].

In addition to alterations in immune defenses, the immunologic changes associated with pregnancy also affect the pathogenesis of autoimmune disease, which can alter the symptom profile of pregnant patients. The shift in cytokine profile from a Th1- to a Th2-mediated phenotype improves inflammatory-type autoimmune disease, while humorally-mediated autoimmune disease is exacerbated [5]. Obesity and diabetes during pregnancy also contribute to the development of complications such as sleep deprivation or increased susceptibility to parasitic infections, which can impact women’s immunity and vaccine effectiveness [10,11,12].

With respect to COVID-19 infection, it was observed early in the pandemic that obese and diabetic patients were more susceptible to infection, had more severe courses of illness, and had worse prognoses than normoglycemic patients. While subsequent research has confirmed this, the molecular mechanisms have not been fully elucidated, though the scientific community has developed theoretical explanations based on pre-established biomedical understanding and scientific studies of similar disease (e.g., severe acute respiratory syndrome). Viral infection decreases angiotensin converting enzyme-2 (ACE2) expression, which leads to increased Angiotensin II activity and the subsequently exaggerated immune response observed in COVID-19 patients. Viral binding to ACE2 receptors down regulates their expression on alveolar epithelial cells, which, after enhancing the angiontensin II-receptor expression, activates the host’s adaptive immune response; this adaptive immune response is amplified—the so-called “cytokine storm”—and involves the release of many inflammatory markers (e.g., IL-1β, IL-4, IL-10, monocyte chemotactic peptide-1 (MCP-1), interferon γ (IFN-γ). This “cytokine storm” is both a hyper-inflammatory and hyper-coagulatory response, which disrupts endothelial cell integrity and thus hinders the alveolar–capillary barrier, resulting in severe hypoxemia [13,14].

Due to altered maternal immunity, as well as the adaptive respiratory, cardiovascular, and hematologic changes outlined above, much concern has been raised about the potential vulnerability of pregnant women to COVID-19 infection. While there is little evidence regarding the impact of COVID-19 in first-trimester pregnant patients, seasonal influenza has been associated with higher rates of early miscarriage. Based on evidence of other viruses affecting third-trimester patients, researchers have theorized that COVID-19 infection may increase the likelihood of various adverse pregnancy outcomes, including fetal growth restriction, preterm delivery, and perinatal mortality. Early data have been reassuring, however, and most studies have observed no greater risk of developing severe COVID-19 among pregnant women than among the general population [15]. While GDM was not independently associated with poor COVID-19 prognosis, obese patients with insulin-dependent GDM showed an increased risk of developing severe COVID-19; furthermore, adverse obstetric outcomes were more common in COVID-19 patients with newly or recently diagnosed GDM, suggesting that well-controlled diabetes confers a better COVID-19 prognosis than unregulated GDM [16].

Vaccine effectiveness is a measure of how well a given vaccination protects people against poor health outcomes due to the disease in question. It is typically measured by comparing the frequency of adverse health outcomes in vaccinated people with that of unvaccinated people; adverse health outcomes may include asymptomatic infection, disease attacks, hospitalizations and deaths due to the disease, serious adverse events due to vaccination, and vaccine reactogenicity [17].

Due to the immunologic changes associated with obesity and DM, this particular patient population appears to be less responsive to some vaccines [18,19], including COVID-19 [20]. The immunologic adaptations of pregnancy also appear to reduce vaccine effectiveness in general; though studies have established efficacy of the COVID-19 vaccine in pregnant women [21], the low numbers of vaccinated pregnant women have made this topic difficult to research [22].

The current study aims to review the immunologic complications observed in pregnant patients with DM and/or obesity, and to further elucidate the biomolecular mechanisms associated with the physiological state and subsequent obstetric complications. The authors also aim to review vaccine effectiveness within this patient population—both in general and with respect to COVID-19 vaccination.

## 2. Obstetric Complications in Women with Obesity and Diabetes Mellitus

According to the World Health Organization, obesity is defined as a body mass index (BMI) above 30 kg/m^2^ and is becoming a non-infectious epidemic of the 21st century, reaching epidemic proportions in both developing and developed countries [23,24]. The prevalence of obesity in women of reproductive age is rising, and it is becoming a common risk factor in obstetrics [25]. Overweight or obese women are more prone to experiencing excessive gestational weight gain (GWG) during pregnancy compared to women with normal weight. GWG is an independent risk factor for the progression of cardio-metabolic complications in both mothers and offspring [26]. Diabetes is a condition characterized by increased blood glucose levels and glucose intolerance, detected for the first-time during pregnancy (GDM) or before conception (pre-gestational diabetes) [27]. GDM is a significant health concern occurring in 2–6% of pregnancies, and in the presence of pre-pregnancy obesity, it rises to 17% in industrialized countries [25]. The detection of GDM includes the oral glucose tolerance test (OGTT) for every healthy pregnant woman at around 24 to 28 weeks of pregnancy, and sooner for women in high-risk groups [28,29].

Physiologically, during pregnancy, the levels of hormones such as placental lactogen, progesterone, estrogen, and cortisol rise, leading to an increase in insulin resistance. Furthermore, evidence suggests that adipose tissue, through the production of adipocytokines—particularly TNF-α—may contribute to a decrease in insulin sensitivity [30]. Obesity is one of the main causes of metabolic dysfunction during pregnancy, particularly GDM and T2DM [28]. The risk of developing hypertensive disorders, such as pregnancy-associated hypertension, is positively correlated with GDM due to similar underlying pathophysiology [31]. Moreover, women diagnosed with GDM are at risk of developing dyslipidemia, considering the higher levels of triglycerides and lower levels of HDL-cholesterol detected at the end of the second trimester [32].

Maternal obesity and diabetes in pregnancy have independent and interconnected impacts on maternal and neonatal outcomes. Both conditions are associated with in utero complications, including stillbirth, fetal macrosomia, being large for gestational age (LGA), preterm birth, and post-delivery complications such as perinatal death, neonatal death, or infant mortality. These conditions are correlated with an increased risk of postpartum depression, stroke, the development of cardiometabolic diseases, or T2DM in the future [26,33]. According to Egan et al., the risk of developing T2DM within 15 years of delivery was found to be 30% for women with GDM at an optimal weight, compared to 70% for obese women with GDM [28].

Obstetric-related complications include shoulder dystocia, prolonged labor, instrumental deliveries, and cesarean section. Furthermore, overweight and obesity have been associated with an increased risk of anesthesia-related complications, such as higher rates of failed initial regional epidural and an increased risk of unsuccessful intubation during cesarean section [34]. Obesity may also cause puerperal complications, such as postpartum hemorrhage, venous thromboembolism, extended wound healing, or infections [35]. Robinson et al. discovered that obese women, compared to non-obese women, had a higher risk of wound infection, ranging from 1.7 times (OR 1.67, CI 1.38–2.00) in moderate obesity to 4.8 times (OR 4.79, CI 3.30–6.95) in severe obesity. Furthermore, obesity was identified as an independent risk factor for wound infections, even when prophylactic antibiotics were used [34]. Suggested wound dressing options for this group of patients include low-adherence, hydrocolloid, and hydrogel dressings. Patients should also regularly inspect the wound area and keep it clean and dry [36].

Both obesity and diabetes are associated with chronic low-grade inflammation that can lead to an overactive immune response. Shannon M. Heitritter et al. presented research showing that women with GDM had higher levels of inflammation biomarkers, such as CRP, IL-6, and PAI, compared to the control group [37]. Additionally, adipose tissue, apart from adipocytes, contains numerous fibroblasts and immune cells, such as mast cells, leukocytes, and macrophages [38]. Maternal overweight or obesity and GDM lead to metabolic dysfunction characterized by hypertriglyceridemia, hypercholesterolemia, hyperglycemia, insulin resistance, and as a complication, chronic oxidative stress and a low-grade inflammatory state [39].

Gynecological care and health examinations for women with obesity, overweight, or pre-gestational diabetes prior to pregnancy should focus on weight loss and achieving normalized glucose levels before conception [26]. However, interventions including non-pharmacological methods (proper diet, physical activity) and pharmacological methods (Figure 1) should be conducted throughout the entire lifespan to improve the metabolic health of the mothers and their offspring and minimize pregnancy-related complications. This applies not only to the antenatal, prenatal, or postnatal periods [33].

## 3. Mechanisms Influencing the Immune Response in Diabetic and Obese Pregnant Patients

In pregnancy the maternal organism develops adaptive mechanisms within the immune system for pregnancy progress. Nevertheless, those adaptations may vary depending on maternal health status and potential occurrence of complications. Among the main phenomena typically observed in physiological, healthy gestation processes are i.e., increased ability to secrete IL-8, IL-12, and IL-1beta. On the other hand, suppression of TNF alpha and production of IFN gamma by NK cells can be observed as well. Additionally, blood cells gain stronger properties to enhance chemotaxis and immunoglobulin opsonization. Moreover, progesterone and estrogen show the ability to regulate cytokine synthesis in pregnancy [40]. The crucial role in pregnancy maintaining is also regulation of M1 and M2 macrophages rate within decidua and here Treg involvement status is essential for maternal–fetal immune tolerance [41].

Obesity is a condition that impairs the immune system and affects leukocyte status and cell-mediated immunity. Chronic inflammation is a consequence of the correlation between the immune system and adipose tissue [42]. Perez de Herida et al. investigated mechanisms through which obesity alters the immune response in general. They identified four initial mechanisms: an increase in leptin secretion and a decrease in adiponectin (ADP) secretion, inflammation caused by non-esterified fatty acids, for example, through toll-like receptor modulation, endoplasmic reticulum stress resulting from adipocyte expansion, and hypoxia originating from hypertrophic fat tissue leading to overexpression of genes responsible for inflammation and immune cell activation. These theories have been found to be accurate in the context of pregnant patients and are currently undergoing further investigation [42].

Moreover, obesity is a significant risk factor for insulin resistance and the development of T2DM. Metabolic dysregulation in obesity results in immune activation as it involves various immune cells. The increased amount of adipose tissue promotes a chronic low-grade inflammation process. Fat tissue primarily includes numerous types of immune cells, such as macrophages, neutrophils, eosinophils, dendritic cells (DCs), mast cells, innate lymphoid cells (ILCs), and natural killer (NK) cells. On the other hand, adaptive immune cells include T and B lymphocytes [43]. Hypertrophic adipocytes can increase the production of various molecules, including TNF, interleukin (IL)-6, monocyte chemoattractant protein 1 (MCP-1), interleukin 1 (IL-1), and interleukin 8 (IL-8). Leptin, as a proinflammatory adipokine, is overexpressed in obese pregnant patients and also contributes to insulin resistance [43]. Leptin induces Th1 cytokine production, while adiponectin expression, which diminishes inflammatory response, is decreased in patients with obesity [44,45,46,47]. Decreased insulin sensitivity correlates with immune modifications during pregnancy, such as increased circulating TNF-alpha and IL-6, which are believed to trigger obesity-associated metabolic inflammation [48]. In general, chronic inflammation is believed to be the main factor contributing to the occurrence of insulin resistance [49]. Numerous studies have found biologically active lipids (such as long-chain cholesterol acyl transferase (acyl-CoA), ceramides, and diacylglycerols) in adipose tissue responsible for the production of adipokines and proinflammatory cytokines [50]. In their study, Liu et al. focused on the role of regulatory T (Treg) cells in obesity-associated insulin resistance (IR). Additionally, dendritic cells, eosinophils, and type 2 innate lymphoid cells (ILC2s) are involved in the regulation of adipose tissue homeostasis by supporting Treg development [51]. Treg cells derived from adipose tissue differ from those found in lymph nodes and spleen. Yuan et al. discovered that Tregs, IL-10, and TGF-beta levels are decreased in patients affected by T2DM and obesity [52]. IL-10, primarily known as a crucial inflammation suppressor, is secreted by decidual macrophages (DMs), decidual natural killer cells (dNKs), and decidual dendritic cells (dDCs). In pregnancy, IL-10 also regulates DMs differentiation to modulate and trigger dNK cell toxicity. Furthermore, IL-10 maintains tissue sensitivity to insulin [41,53].

Interleukin 4 (IL-4), interleukin 13 (IL-13), interleukin 5 (IL-5), and the alarmin interleukin 33 (IL-33) are known mediators of the type 2 immune response in adipose tissue. Generally, the type 2 response is responsible for minimizing inflammation in lean adipose tissue. However, with increased adiposity, molecules such as monocyte chemoattractant protein-1 (MCP-1), interleukin 6 (IL-6), TNF, leptin, and adipocyte cell death recruit macrophages, neutrophils, and CD8 T cells, thus exacerbating inflammation [43].

Monocytes play a crucial role in immune response and are known for their ability to secrete cytokines. In the context of GDM, they have emerged as essential agents in the immune system, potentially influencing the inflammatory processes associated with this condition [54]. Han et al. observed that IL-6 has dual properties. IL-6 secreted in myeloid cells are adipose tissue macrophage inhibitors, whereas those derived from adipocytes and myocytes induce ATM recruitment [55].

In comparison to pregnant patients without GDM, those with GDM exhibit distinct immune profiles. Specifically, they display lower levels of M2 macrophage markers and IL-10, which are associated with anti-inflammatory responses. On the other hand, GDM patients show elevated levels of M1 macrophage markers, TNF-α and IL-6, which are indicative of pro-inflammatory activity. These differences in immune markers suggest a shift towards a more inflammatory state in GDM [54]. Researchers, such as Huang et al., have put forward the hypothesis that the reduced monocyte count observed in GDM could be attributed to an imbalance between decreased anti-inflammatory monocyte subsets and elevated pro-inflammatory monocyte subsets in the peripheral blood. This imbalance may contribute to a chronic inflammatory state in GDM patients, perpetuating the immune dysregulation observed [54].

Patients with hyperglycemia first detected in pregnancy (HFDP) exhibit elevated levels of neutrophils and monocytes, which contribute to the activation of the inflammatory response. The presence of hyperglycemia during pregnancy can lead to dysregulated immune responses and inflammatory processes. Researchers have focused on evaluating specific ratios, namely the neutrophil-to-lymphocyte ratio (NLR) and monocyte-to-lymphocyte ratio (MLR), to better understand the relationship between inflammation and glucose regulation.

The NLR, which represents the ratio of neutrophils to lymphocytes in the bloodstream, has emerged as a potential marker of systemic inflammation. In patients with HFDP, an elevated NLR suggests a heightened inflammatory state. Moreover, studies have shown that the NLR is associated with impaired glucose regulation in T2DM, indicating a potential link between inflammation and glucose metabolism dysfunction.

Similarly, the MLR, calculated by dividing the number of monocytes by the number of lymphocytes, has garnered attention as a potential indicator of inflammation and immune system dysregulation. Elevated MLR levels in patients with HFDP further support the presence of ongoing inflammation [56].

Wender-Ozegowska et al., in their study, examined the concentrations of chemokines in peripheral blood during the first trimester of pregnancy in women with type 1 diabetes mellitus (T1DM). The results demonstrated that in the tested group, levels of matrix metalloproteinase-9 (MMP-9) and MCP-1 were higher, while levels of IFN-γ-inducible Protein 10 kDa (IP-10 or CXCL10) and cytokines regulated upon activation normal T cell expressed and secreted (RANTES) were lower compared to the control group consisting of non-diabetic individuals [57].

Another study aimed to analyze the immunological features of pregnant patients with T1DM. The report reveals a significant elevation in the white blood cell (WBC) count, and unlike women without T1DM, their level of lymphocytes during pregnancy did not decrease. Additionally, it was observed that pregnant women with T1DM had an elevated Th1/Th2 ratio and increased NLp46 expression on NK cells compared to patients without comorbidities. Furthermore, the mean fluorescence intensity (MFI) of MHC-II on intermediate and non-classical monocytes was higher in pregnancies complicated by T1DM. One theory suggests that these differences may arise from the presence of autoimmune disease, which modulates the response to various pregnancy factors. Groen et al. confirm that pregnancy in diabetic women is characterized by an up regulation of the Th1 immune response and monocyte activation [58]. According to the study by Sen et al., the proportion of CD8 T cells (CD3/CD8) was significantly lower in obese patients compared to lean patients. Body mass index (BMI) and CD8+ cell count were also found to be negatively correlated. Furthermore, it was observed that obese patients had a smaller percentage of NKT cells (CD3+/CD314+) and a higher percentage of B cells (CD19+) compared to lean patients. Following stimulation with PMA (phorbol myristate acetate) and ionomycin, lower levels of intracellular INF-α and INF-γ were observed in CD4+ and CD8+ cells of obese patients. Sen et al. also reported impaired proliferation in response to anti-CD3/CD28 in obese patients. This group also exhibited higher levels of c-reactive protein (CRP) and oxidized-to-reduced glutathione. Additionally, the authors found correlations between leptin and adiponectin levels and CD8+ cell count, B cell count, as well as TNF-α and IFN-γ production from CD8+ cells. 

Adipose tissue, functioning as an endocrine organ, has the capability to secrete chemokines, adipokines, and cytokines, and may play a role in modulating the immune response of pregnant individuals. The authors hypothesize that progesterone, an immunosuppressive hormone accumulating in adipose tissue, could impair T cell function [59]. Wei et al. also demonstrated that elevated levels of TNF alpha and decreased ADP have been observed in the serum of pregnant patients with GDM [49]. Macrophages migrate to the adipose tissue in response to the secretion of MCP-1/CCL2 and leukotriene B4 by adipocytes, which are induced by circulating triglycerides (TG) and leptin secretion. These adipose tissue macrophages (ATMs) contribute to inflammation by secreting TNF-α, IL-6, and IL-1β, leading to the activation of the nuclear factor-κB (NF-κB) signaling pathway [60]. Additionally, IL-1β can decrease insulin production and triggers beta cell apoptosis [61].

Pregnant patients with higher pre-gravid BMI were found to have higher levels of IL-6 during pregnancy. Pre-gravid obesity is associated with elevated levels of C-C Motif Chemokine Ligand 2 (CCL2) and CXCL8, which are recruiting factors in the blood during pregnancy. Furthermore, studies have shown that pre-gravid obesity also influences the levels of granulocyte-macrophage colony-stimulating factor (GM-CSF) and fibroblast growth factor 2 (FGF-2) [62].

Another study demonstrated that murine models with obesity induced by a high-fat diet exhibited up-regulated T lymphocytes in lymph nodes and peripheral blood during normal pregnancy (NP). Additionally, it was observed that in murine models prone to abortion, obesity resulted in increased numbers of decidual natural killer cells (dNK cells) and heightened activity of the CD11+ bCD27+ subset [63].

Obesity and diabetes are closely linked to chronic inflammation and their association with gut microflora. The human microbiota consists of approximately 100 trillion organisms, primarily bacteria [64]. The most common groups of bacteria found in the microbiota are Firmicutes, Bacteroidetes, Actinobacteria, and Proteobacteria. Overweight patients’ gut microbiota is characterized by increased amount of Firmicutes, Fecal cocci, Streptococcus, and Actinomycetes species [64,65,66]. There are numerous reports proving that gut microbiota is involved in the development of GDM. The pivotal role of gut microbiota in this case lies in its ability to regulate insulin resistance and inflammation. Pregnancy is associated with significant hormonal, metabolic, and immunological changes, and these profound modifications also affect the gut microbiota. These changes include alterations in bacterial flora and a decrease in richness (α-diversity) throughout pregnancy [67,68].

Based on the available data, it can be inferred that dysbiosis of the gut microbiota and GDM are closely linked, similar to the association between gut microbiota and T2DM. The gut microbiota could potentially serve as a risk marker for the development of GDM. Certain species, such as Ruminococcaceae, Parabacteroides distasonis, and Prevotella, which interact with metabolic pathways involved in carbohydrate metabolism and insulin signaling, are more prevalent in GDM patients [67].

Additionally, the GDM group was found to have increased levels of Klebsiella variicola, Ruminococcus, Eubacterium, Collinsella, Rothia, Desulfovibrio, Actinobacteria, and Firmicutes. On the other hand, there was a reduced gut abundance of Methanobrevibacter smithii, Alistipes species, Bifidobacterium species, Eubacterium species, Akkermansia, Bacteroides, Parabacteroides, Roseburia, and Dialister. It is interesting to note that the elevation of Roseburia and Akkermansia muciniphila is observed not only in GDM but also in T2DM. Proteobacteria, which are Gram-negative bacteria, are widely associated with inflammation in patients with T2DM and GDM. However, there are inconsistent reports on this topic, and some studies even report a negative correlation between weight gain and proteobacteria density. It is worth mentioning that patients with GDM are seven times more likely to develop T2DM in the future [67,68].

Koren et al. discovered that gut microbiota dysbiosis during the third trimester is associated with low-grade inflammation involving IL-1, IL-2, IL-5, and IL-6, as well as insulin resistance and hyperglycemia [69]. Weight gain during pregnancy promotes an increase in populations of Bacteroides species, Staphylococcus, Enterobacteriaceae, and Escherichia coli, while reducing the populations of Bifidobacterium and Akkermansia. Importantly, Akkermansia is associated with lower insulin sensitivity in patients with insulin resistance, and the Firmicutes-to-Bacteroidetes ratio can be elevated, with a decrease in the number of butyrate-producing bacteria, including Roseburia or Faecalibacterium prausnitzii. Faecalibacterium is known to be an anti-inflammatory agent and is inversely associated with fasting blood sugar levels [67,68]. Elevated concentrations of pathobionts have the properties to activate pathogenic inflammatory responses and harm the host. Provotella bacteria are known as branched short-chain amino acid synthesizers, which can alter insulin sensitivity and lead to an insulin-resistant state [70]. The presence of LPS in the Bacteroides cell wall, by binding to toll-like receptor 4 (TLR4), can also alter signaling pathways and eventually lead to insulin resistance [70].

Clearly, diet has a strong impact on gut microbiota, and dietary choices are also linked to the occurrence of GDM and T2D. Methylamines, which are present in eggs, meat, and salt-water fish, can be oxidized by gut microbes to trimethylamine N-oxide (TMAO). According to reports, high plasma concentrations of TMAO in early and mid-pregnancy are positively correlated with the risk of developing GDM [68,71]. Elevated TMAO levels are also associated with an increased risk of developing T2D [68,72]. Higher fat intake could lead to an increase in pro-inflammatory bacteria, subsequently resulting in insulin resistance and a reduction in bacterial richness [68].

## 4. COVID-19 in Pregnant Women with Obesity and Diabetes

In December 2019, the first cases of infection caused by a new virus—severe acute respiratory syndrome coronavirus 2 (SARS-CoV-2)—were identified in Wuhan, China [73]. After a few months, the disease was recognized as a pandemic by the World Health Organization (WHO). The new disease has been named as coronavirus disease 2019 (COVID-19) [74]. The lockdowns implemented during the pandemic have led to unhealthy habits, such as improper diets and reduced physical activity among the population, including pregnant women. Consequently, there has been an increased prevalence of insulin resistance, body fat, pro-inflammatory cytokines, and diseases such as T2DM, GDM, and obesity [75]. It is worth noting that even healthy pregnant women are considered particularly susceptible to COVID-19 due to cellular immunity deficiencies which increase vulnerability to viral infections [20]. Studies have shown that pregnant women are at risk of experiencing a severe course of COVID-19. In such cases of infection, preterm delivery and mortality can occur. There is also an increased likelihood of hospitalization, requiring mechanical ventilation, and admission to the intensive care unit [76]. Pregnant women with pre-existing conditions such as obesity, hypertension, and respiratory tract diseases, particularly asthma, are at the highest risk of developing symptomatic infection [20]. Additionally, pregnant women with COVID-19 are more likely to experience preterm labor (before 37 weeks of pregnancy) and have higher rates of cesarean sections and neonatal intensive care unit admissions, while the rates of intrauterine and neonatal mortality remain low [20,77]. Elevated levels of IL-8 have been observed in the blood of women infected with SARS-CoV-2 and their babies. Therefore, IL-8 may be used as a biomarker for predicting disease severity and patient survival in such cases [76].

The global prevalence of overweight and obesity among pregnant women is increasing, and viral infections in this population can significantly impact the health of both mother and child. In pregnant women with an elevated body mass index (BMI ≥ 25.0 kg/m^2^), SARS-CoV-2 infection may be associated with placental pathologies such as chronic inflammation, fetal vascular malperfusion, maternal vascular malperfusion, fibrinoid lesions, and myxoid lesions [78].

Table 1 illustrates the consequences of these pathologies for both the mother and the offspring.

In the context of obesity, elevated levels of pro-inflammatory cytokines are observed, and in pregnancies complicated by obesity, an increase in their concentration in the placenta is observed [79].

In this situation, viral infection can start the process of destroying the cells of the placenta and its dysfunction in terms of nutrient transport, which can have a negative impact on further fetal development [80].

Obesity and T2DM are classified as civilization diseases due to their high occurrence in the global population. As the prevalence of these diseases in both non-pregnant individuals and pregnant women increases the risk of COVID-19 infection, the INTERCOVID study aimed to investigate the associations between COVID-19 diagnosis and pre-existing diabetes, high BMI, or GDM in pregnant women [81]. It was found that pregnant women with pre-diagnosed diabetes had a two-fold higher risk of COVID-19, while women who were overweight or obese, as well as women with diagnosed GDM, had an approximately 20% higher risk of infection than pregnant women in the control group [81]. The strongest association with COVID-19 diagnosis was observed in women with previously diagnosed diabetes who were overweight or obese, followed by women with previously diagnosed diabetes and normal weight [81]. GDM is a risk factor for severe COVID-19 and hospitalization especially in women with obesity. However, good glycemic control may have a beneficial effect on clinical outcomes in patients with GDM and COVID-19 [75]. The interactions between COVID-19, diabetes and pregnancy are complex. SARS-CoV-2 may determine hyperglycemia and diabetes due to its interaction with angiotensin-converting enzyme 2 (ACE2) receptors and the resulting damage to pancreatic islet β-cells [75]. The mechanisms that occur after the viral entry into pancreatic islet cells in pregnant women are shown in Figure 2.

ACE2 receptors are present in most of the cells, including pancreatic islet cells, over-expressed in patients with diabetic mellitus. SARS-CoV-2 interaction with ACE2 receptors on the pancreatic islet cells in pregnant women leads to the activation of ACE2 pathway. It may cause acute ß-cell dysfunction, and in consequence a hyperglycemic state. It can worsen the course of DM and even promote the disease de novo. Chronic hyperglycemia is associated with increased viral replication and with suppression of antiviral immune response in pregnant tissues [75].

It is also important that COVID-19 is associated with a state of hypercoagulation also present during pregnancy in healthy women, thus increasing the maternal thromboembolic risk already associated with obesity and diabetes [82].

Due to the increased risk of SARS-CoV-2 infection during pregnancy, especially in women with obesity and diabetes, the scientific community considers it a priority for pregnant women to receive the COVID-19 vaccine [77]. Vaccination provides protection to both the mother and the baby through passive immunization. Vaccine-induced IgG immune antibodies are transferred transplacentally during pregnancy and through breast milk [21,83]. Data from the 2022 study showed the effectiveness of COVID-19 vaccination in preventing infant hospitalization at 32% before 20 weeks of pregnancy and at 80% after 20 weeks [84]. Despite concerns about the safety of mRNA vaccines against COVID-19 during pregnancy and lactation, several studies demonstrated their safety [21,84,85,86]. The vaccines showed efficacy in terms of increasing SARS-CoV-2 antibody levels, with only minimal adverse effects, such as 8% risk of preterm births and local symptoms after vaccination [21]. A trial involving 127 pregnant women who received the vaccine during pregnancy showed no negative impact on perinatal outcomes compared to unvaccinated pregnant women [85]. The study also revealed that pregnant women with diabetes are more likely to receive the vaccine than healthy pregnant women, while factors associated with lower vaccination rates included low maternal socioeconomic status, maternal age under 30, and Afro-Caribbean origin [85]. Also important is the fact that vaccination has no effect on the occurrence of pathological changes in the placenta [86]. Despite the growing approval of COVID-19 vaccination in the general population, acceptance of vaccination remains low among pregnant women. The 2021 study revealed that out of 300 pregnant women, only 111 of them (37%) were willing to be vaccinated against COVID-19 [22]. The most common reasons for refusing vaccination reported by the surveyed women included lack of data on the safety of COVID-19 vaccination in pregnancy and the possibility of fetal harm, while these concerns were more common in women with high-risk pregnancy [22]. Figure 3 presents the most common concerns of pregnant women related to COVID-19 vaccination, divided into high- and low-risk pregnancies.

Considering the risk of severe COVID-19, the benefits of vaccinating pregnant women outweigh the potential risks, especially as vaccines against COVID-19 do not contain the live, attenuated virus. It is recommended that women of reproductive age be advised to receive the vaccine when planning pregnancy and educated on the effectiveness and safety of COVID-19 vaccination.

## 5. The Effectiveness of Vaccinations in Pregnant Women with Obesity and Diabetes

Maternal vaccination (MV) has been scientifically proven to be beneficial in protecting women, their fetuses, newborns, and young children from various conditions. Immunizing women during pregnancy helps protect the mother from illnesses to which she may be particularly vulnerable, while also shielding the unborn baby from congenital infections and other adverse effects of maternal infection. Moreover, through the placental transfer of neutralizing immunoglobulin G (IgG) antibodies and/or secretory immunoglobulin A (IgA) antibodies in the mother’s breast milk, maternal vaccination may be used with the primary aim of protecting the developing fetus and infant from infection during the first months of life [87,88].

Vaccines containing dead or inactivated viruses, protein components or toxoids, and conjugate vaccines (such as protein-toxoid, peptide-protein, and protein-protein) are all considered safe for use during pregnancy. Considering the hypothetical risk of congenital infection and the potential increase in the risk of miscarriage, live vaccines are generally not recommended for pregnant women, with a few exceptions. In some instances, when the potential risks to the fetus are believed to be significantly outweighed by the risk to the mother, live vaccines may be considered [87].

Currently, both international and national health organizations recommend the seasonal influenza vaccine and the tetanus toxoid, diphtheria toxoid, and acellular pertussis (Tdap) vaccine for pregnant women [89]. Since 2004 and 2012, respectively, The Advisory Committee on Immunization Practices (ACIP) has advised the use of these vaccines for every pregnancy [90]. Tdap vaccination in a single dose is recommended between 27 and 36 weeks of gestation, but it can be administered earlier if necessary. Bearing in mind that various studies have shown that pregnant women are at a greater risk of severe disease and death from seasonal influenza than non-pregnant women, the influenza vaccine should be administered before the start of flu season, regardless of the stage of pregnancy. Furthermore, a number of vaccines—including those that guard against the cytomegalovirus (CMV), respiratory syncytial virus (RSV), and group B streptococcus (GBS)—are still being studied and may soon be approved for use during pregnancy [87].

Obesity has a clinically significant impact on immunity and pathogen defense. Excessive body weight leads to the alteration of leukocyte development, phenotypes, and activity, disruption of lymphoid tissue integrity, and the coordination of innate and adaptive immune responses. This undoubtedly has a negative effect on vaccine efficacy [91]. Numerous studies have shown higher rates of vaccine failure and complications from infection in obese patients [92], however they did not apply to pregnant women and were most commonly related to the hepatitis B vaccine. Additionally, reactions to the tetanus and rabies vaccines were found to be corresponding inversely with body weight [93,94].

A prospective observational study was carried out by Sheridan et al. to assess the immune response to the 2009–2010 inactivated trivalent influenza vaccine (TIV) in subjects with healthy weight, overweight, and obesity. Obese individuals have shown decreased IgG production 12 months after vaccination, as well as reduced CD69, IFN-γ, and granzyme B expression in CD8+ T cells [95]. In contrast, another study aimed at determining the association between obesity and serologic response to influenza vaccination in older adults found no consistent relationship between antibody response and obesity [96].

According to several studies, the efficacy of influenza vaccines is similar in the group of people with diabetes and in healthy, non-diabetic people [97,98,99]. Diabetic adults who receive the influenza vaccine can benefit from its effectiveness and safety. The vaccine not only helps protect them from influenza but also contributes to lowering death and hospitalization rates [100].

Currently, there is a lack of information regarding the efficacy of vaccines in pregnant women with obesity and diabetes. Considering that both pregnancy and obesity can lead to impaired immune systems, it is possible that the effectiveness of vaccines may be decreased in these individuals. At the same time, vaccinations are crucial and essential especially for patient groups that are more susceptible to serious diseases.

In a retrospective cohort study, involving 34,701 pregnant women delivering between April 2012 and December 2013, Regan et al. [101] evaluated the effectiveness of seasonal trivalent influenza vaccination against hospital-attended acute respiratory infections. A total of 6280 included subjects suffered from obesity, 229 had pre-existing diabetes, and 2746 manifested GDM. The level of vaccination coverage accounted for 8.7% of the whole examined population and was 9.9%, 13.1%, and 11.2% in obese patients, women with pre-existing diabetes and GDM, respectively. Overall, results showed that vaccinated women were less likely to visit an emergency department (ED) during pregnancy for an acute respiratory illness (ARI) and the likelihood of ARI-related hospitalization was also lower among this group. Moreover, adjusted analyses revealed that obese women were more likely than non-obese women to visit a hospital ED for an ARI during influenza season [101]. In conclusion, the study found that influenza MV is effective not only in the general population of pregnant women but also in the group of pregnant patients who are obese and diabetic.

The review article by Karlsson et al. focused on the impact of pregnancy and obesity on influenza virus [A(H1N1)pdm09 virus] infection, covered the aspect of prophylaxis as well [102].

Obesity and pregnancy have both been linked to the development of severe influenza virus-related illness. Compared to non-pregnant and non-obese populations, these groups have been found to be more prone to experiencing respiratory problems, such as pulmonary distress. Hospitalization and mortality rates were disproportionately higher among obese and pregnant patients. Due to the increased severity of infection, pregnant women should be a high-priority group for vaccination. However, the overall efficacy of the vaccine remained uncertain and difficult to determine. Limitations, like fairly low vaccine coverage (VC) in pregnant women (e.g., 12% in France; 10.3% in Western Australia), made it difficult to assess vaccine effectiveness [102].

Insufficient vaccine coverage seems to be the main challenge during vaccination efficacy assessment in pregnant women with obesity and diabetes.

Fernández-Cano et al. summed up available data from between 2015 and 2018 to determine the coverage of both, recommended during pregnancy vaccinations (against seasonal influenza and TdaP) in the health area of Catalonia, Spain. The analysis comprised 36,032 data of pregnant women in total, including individuals with pregnancy risk conditions. Obesity was the most frequent risk factor and accounted for 12.5% of subjects. Pregnant women suffering from T1DM and T2DM constituted 0.5% of those included in the study. What is concerning is that more than half of them (56%) did not report any vaccination, which makes diabetics the least vaccinated group of all those at risk. Researchers noted an increasing trend in coverage with the tetanus toxoid, diphtheria toxoid, and acellular pertussis vaccine (49.8% in 2016 and 79.4% in 2018); however, the most significant variable associated with a lack of MV against pertussis was diabetes (39% less likelihood of vaccination). The coverage of vaccination against influenza followed a decreasing trend (11.9% in 2015 and 6.8% in 2018). Obese women proved to be the least likely to vaccinate. The association between obesity and a lower level of education could be one of the possible explanations for this state of affairs [90].

The assumption that the influenza vaccine is ineffective and unsafe for the fetus is one of the justifications given by pregnant women for not receiving it. This highlights the demand for better education for pregnant women [90].

A study conducted in July 2010 and July 2011, involving 199 new mothers and 240 new mothers, respectively, aimed to improve influenza vaccination coverage during pregnancy [103]. The study concluded that there is a need to increase awareness of the benefits of vaccination. Taking actions such as conducting lectures for healthcare professionals and providing information brochures to patients had positive effects. Thanks to the educational program, an increase in influenza vaccination from 30% in 2010 to 40% in 2011 was noted [103].

## 6. Conclusions

Maternal obesity and diabetes are associated with numerous risks for both the mother and child, both during pregnancy and postpartum. These conditions can alter the immune profiles of pregnant women, with differences observed between preexisting diseases and those acquired during pregnancy. Antenatal care should place greater emphasis on weight control and achieving normal blood sugar levels to prevent perinatal and long-term complications.

Pregnant women are more likely to experience severe symptoms of COVID-19, in addition to obstetric complications such as preterm birth. Co-morbidities like obesity, diabetes, and hypertension are associated with a higher risk of infection and poorer prognosis, and COVID-19 infection in obese mothers may be linked to various placental pathologies.

Studies have demonstrated the efficacy of COVID-19 vaccination in pregnant women. While the scientific community has concluded that the benefits of COVID-19 vaccination outweigh the minimal risks, adherence to vaccine recommendations remains low among pregnant women. Since pregnant women and their fetuses are particularly vulnerable to severe COVID-19 symptoms, it is crucial to prioritize patient education and outreach to this at-risk group.

## Figures and Tables

**Figure 1 vaccines-11-01247-f001:**
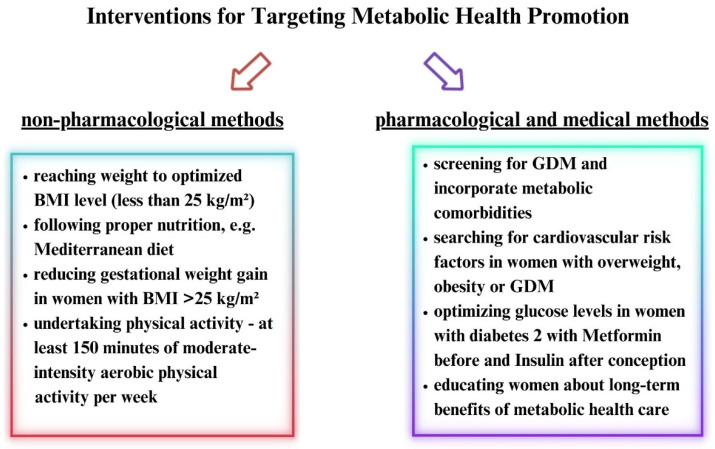
Summary of non-pharmacological and pharmacological or medical interventions including prenatal, antenatal, and postnatal pregnancy periods for metabolic health promotion [33].

**Figure 2 vaccines-11-01247-f002:**
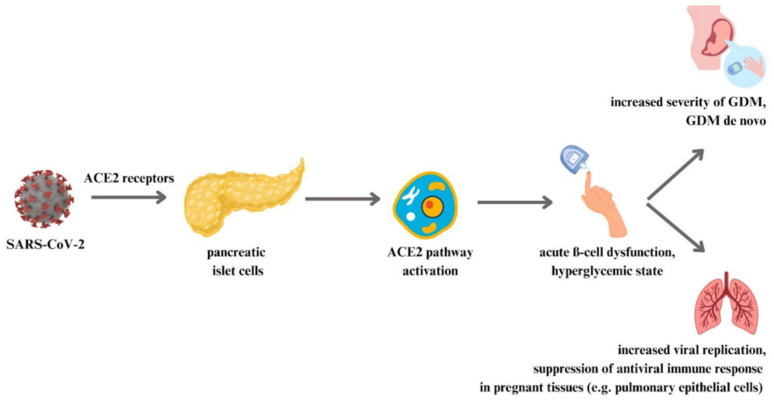
Clinical consequences of viral entry into pancreatic islet cells.

**Figure 3 vaccines-11-01247-f003:**
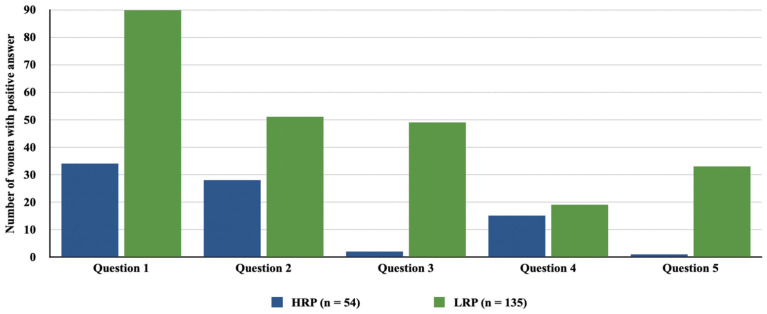
The most common concerns of pregnant women related to COVID-19 vaccination [18]. HRP—high risk pregnancy; LRP—low risk pregnancy. Question 1: Lack of data about COVID-19 vaccine safety in pregnant women. Question 2: Vaccine will harm my baby. Question 3: I do not think the vaccine will work. Question 4: Vaccine will harm my body. Question 5: Family members have hesitancy toward the COVID-19 vaccine.

**Table 1 vaccines-11-01247-t001:** Placental pathologies in overweight and obese pregnant women infected with SARS-CoV-2 [78].

Placental Pathologies	Clinical Relevance for Mother and Child
Chronic placenta inflammation	Preterm labor, intrauterine fetal growth restriction, preeclampsia, fetal demise
Fetal vascular malperfusion	Fetal necrosis, intrauterine fetal growth restriction, neonatal coagulopathies, fetal heart defects, fetal encephalopathy
Maternal vascular malperfusion	Fetal demise
Fibrinoid lesions	Fetal necrosis, neonatal death, recurrent lesions
Myxoid lesions	Preeclampsia, intrauterine fetal growth restriction, premature placental detachment

## Data Availability

Not applicable.

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
