# Peer review of "Impact of Obesity and Diabetes in Pregnant Women on Their Immunity and Vaccination"

_vaccines, 2023, doi:10.3390/vaccines11071247_

Round 1
Reviewer 1 Report
The authors performed an interesting article regarding obesity, diabetes, and pregnancy. The manuscript is informative, however, I would like to suggest that the following issues could be included and discussed in your manuscript:
1) Evidences suggest a strong link between both pregnancy and obesity with sleep deprivation, on the other hand, sleep deprivation is associated with a diminished immune response to vaccination. Please see the suggested references: ---Sleep deprivation and obesity in adults: a brief narrative review. BMJ Open Sport & Exercise Medicine. 2018;4(1):e000392.---Sleep deprivation during pregnancy and maternal and fetal outcomes: is there a relationship?. Sleep Medicine Reviews. 2010;14(2):107-14.
---Focus: Vaccines: Sufficient Sleep, Time of Vaccination, and Vaccine Efficacy: A Systematic Review of the Current Evidence and a Proposal for COVID-19 Vaccination. The Yale Journal of Biology and Medicine. 2022;95(2):221. 2) The authors mentioned TORCH infections. I would like to mention that TORCH co-infections also increase risk of severe sequels for mother and infant: ---ToRCH “co‐infections” are associated with increased risk of abortion in pregnant women. Congenital Anomalies. 2016;56(2):73-8. 3) Both pregnancy and diabetes are among the risk factors for intestinal parasitic infections. On the other hand, parasitic infections are associated with a diminished immune response to vaccination. ---Prevalence of intestinal parasitic infections in patients with diabetes: a systematic review and meta-analysis. International Health. 2023. ihad027.
---Global prevalence of intestinal parasitic infections and associated risk factors in pregnant women: a systematic review and meta-analysis. Transactions of the Royal Society of Tropical Medicine and Hygiene, 115(5), pp.457-470.
---Are pregnant women with chronic helminth infections more susceptible to congenital infections? Frontiers in Immunology. 2014; 12;5:53.
---Do antenatal parasite infections devalue childhood vaccination?. PLoS neglected tropical diseases. 2009;3(5):e442.
---Do parasite infections interfere with immunisation? A review and meta-analysis. Vaccine. 2020;38(35):5582-90.
---The effect of helminth infection on vaccine responses in humans and animal models: A systematic review and meta‐analysis. Parasite Immunology. 2022;44(9):e12939.
---The influence of parasite infections on host immunity to co-infection with other pathogens. Frontiers in immunology. 2018;9:2579.
Author Response
Dear Reviewer 1
Thank you very much for taking the time to read our manuscript and make helpful comments.
The topics and works highlighted by the esteemed Reviewer seem to be very interesting, and we will certainly make use of them in the future with great enthusiasm. However, the highlighted articles do not entirely align with our current manuscript's vision.
Yours faithfully,
Assoc. Prof. Żaneta Kimber-Trojnar, M.D., Ph.D.
Chair and Department of Obstetrics and Perinatology
Medical University of Lublin
Jaczewskiego 8, 20-090 Lublin, Poland
Tel: +48 81 7244 769;
Fax: +48 81 7244 841
E-mail: zkimber@poczta.onet.pl
Reviewer 2 Report
Overall, the paper provides valuable insights into the impact of obesity and diabetes on the immunity and vaccination of pregnant women. However, there are a few areas that need improvement.
1. In the abstract, it would be helpful to provide a brief background to highlight the significance of studying the impact of obesity and diabetes on pregnant women's immunity and vaccination. This will set the context for the readers and make the paper more informative.
2. The content and logic of the introduction section need to be further organized, and the overall text needs to be streamlined to increase readability.
3. The mechanisms underlying reduced immunity in pregnant women with obesity and diabetes are briefly mentioned, but more specific details should be provided. For example, which specific immune cells are affected by insulin resistance? How do pro-inflammatory cytokines and adipokines contribute to the inflammatory response? Elaborating on these mechanisms will enhance the paper's scientific rigor.
4. The role of the gut microbiome in immune dysfunction is mentioned, but it would be beneficial to include recent studies exploring the alterations in gut microbiota composition and their implications for immune function in pregnant women with obesity and diabetes.
5. The importance of further research and the need for effective interventions are mentioned, but no specific intervention suggestions are provided. It would be valuable to offer recommendations for potential research directions and intervention strategies that could improve immune function in pregnant women with obesity and diabetes.
English needs to be polished appropriately to be concise and readable.
Author Response
Dear Reviewer 2,
Overall, the paper provides valuable insights into the impact of obesity and diabetes on the immunity and vaccination of pregnant women. However, there are a few areas that need improvement.
Thank you for your valuable comments, esteemed Reviewer. We made an effort to implement these valuable suggestions, and we hope that, thanks to your insightful observations, the quality of our article has improved.
In the abstract, it would be helpful to provide a brief background to highlight the significance of studying the impact of obesity and diabetes on pregnant women's immunity and vaccination. This will set the context for the readers and make the paper more informative.
In response to your valuable suggestion, we have enhanced the abstract based on the provided guidelines.
The content and logic of the introduction section need to be further organized, and the overall text needs to be streamlined to increase readability.
In accordance with your valuable comment, we have reorganized the introduction.
The mechanisms underlying reduced immunity in pregnant women with obesity and diabetes are briefly mentioned, but more specific details should be provided. For example, which specific immune cells are affected by insulin resistance? How do pro-inflammatory cytokines and adipokines contribute to the inflammatory response? Elaborating on these mechanisms will enhance the paper's scientific rigor.
The esteemed Reviewer was right in pointing out that a detailed discussion of the mechanisms would enhance the scientific rigor of the article. In response to this comment, we have expanded this section of the paper.
The role of the gut microbiome in immune dysfunction is mentioned, but it would be beneficial to include recent studies exploring the alterations in gut microbiota composition and their implications for immune function in pregnant women with obesity and diabetes.
In accordance with the comment, the latest research on changes in gut microbiota composition and their implications for immune function in pregnant women with obesity and diabetes has been incorporated.
The importance of further research and the need for effective interventions are mentioned, but no specific intervention suggestions are provided. It would be valuable to offer recommendations for potential research directions and intervention strategies that could improve immune function in pregnant women with obesity and diabetes.
The article emphasizes the significant role of preconception actions and patient education. Recommendations regarding potential research directions and intervention strategies are a broad topic, and we will gladly address it in future work.
Yours faithfully,
Assoc. Prof. Żaneta Kimber-Trojnar, M.D., Ph.D.
Chair and Department of Obstetrics and Perinatology
Medical University of Lublin
Jaczewskiego 8, 20-090 Lublin, Poland
Tel: +48 81 7244 769;
Fax: +48 81 7244 841
E-mail: zkimber@poczta.onet.pl
Reviewer 3 Report
dear Authors,
In this interesting review is discussed in detail the immuno complications of pregnancy in combination with diabetes and obesity; Most interestingly, in the contexts of COVID vaccination.
I am finding highly interesting this topic and the analysis of vaccination efficiency in specific cluster of population; The Pregnancy is a complex period for the immunity that in combination with diabetes and obesity could make to the phisicians and involved women to refuse the vaccination.
To clarify the recent experience in this novel vaccines in this population is a great contribution for the scientific community.
The introduction and following paragraphs anr well structured; The authors explain in detail the specific markers, mechanisms and physiological stages involved.
the only weakness that can be pointed out is the lack of summary graphs or quantification. It is clear, this review is not a systematic one therefore for further understanding and deeper conclusions the reader should go to each specific reference. This manuscript is an excellent guide for a further analysis by each reader.
Author Response
Dear Reviewer 3,
In this interesting review is discussed in detail the immuno complications of pregnancy in combination with diabetes and obesity; Most interestingly, in the contexts of COVID vaccination.
I am finding highly interesting this topic and the analysis of vaccination efficiency in specific cluster of population; The Pregnancy is a complex period for the immunity that in combination with diabetes and obesity could make to the phisicians and involved women to refuse the vaccination.
To clarify the recent experience in this novel vaccines in this population is a great contribution for the scientific community.
The introduction and following paragraphs anr well structured; The authors explain in detail the specific markers, mechanisms and physiological stages involved.
Thank you very much for such a favorable review. We are delighted with such a positive reception.
the only weakness that can be pointed out is the lack of summary graphs or quantification. It is clear, this review is not a systematic one therefore for further understanding and deeper conclusions the reader should go to each specific reference. This manuscript is an excellent guide for a further analysis by each reader.
According to valuable feedback, we have enriched the article with charts depicting quantitative data.
We would like to take this opportunity to thank the Reviewers and Guest Editor for all the valuable and highly perceptive remarks which have definitely made a substantial contribution to the quality of our paper.
Thank you for considering our manuscript for publication. We appreciate your time and look forward to hearing from you.
Yours faithfully,
Assoc. Prof. Żaneta Kimber-Trojnar, M.D., Ph.D.
Chair and Department of Obstetrics and Perinatology
Medical University of Lublin
Jaczewskiego 8, 20-090 Lublin, Poland
Tel: +48 81 7244 769;
Fax: +48 81 7244 841
E-mail: zkimber@poczta.onet.pl